# Synthesis and Characterization of the Novel *N^ε^*-9-Fluorenylmethoxycarbonyl-l-Lysine *N*-Carboxy Anhydride. Synthesis of Well-Defined Linear and Branched Polypeptides

**DOI:** 10.3390/polym12122819

**Published:** 2020-11-27

**Authors:** Varvara Athanasiou, Pandora Thimi, Melina Liakopoulou, Foteini Arfara, Dimitra Stavroulaki, Iro Kyroglou, Dimitrios Skourtis, Ioanna Stavropoulou, Panagiotis Christakopoulos, Maria Kasimatis, Panagiota G. Fragouli, Hermis Iatrou

**Affiliations:** 1Department of Chemistry, Industrial Chemistry Laboratory, National and Kapodistrian University of Athens, Panepistimiopolis, Zografou, GR-15771 Athens, Greece; bathanas@chem.uoa.gr (V.A.); panthimi@chem.uoa.gr (P.T.); melinalk@chem.uoa.gr (M.L.); afoteini@chem.uoa.gr (F.A.); dimistavrou@chem.uoa.gr (D.S.); irokrg@chem.uoa.gr (I.K.); skourtisd@chem.uoa.gr (D.S.); iostav@chem.uoa.gr (I.S.); panchristak@chem.uoa.gr (P.C.); mariakasim@chem.uoa.gr (M.K.); 2Dyeing, Finishing, Dyestuffs and Advanced Polymers Laboratory, University of West Attica, DIDPE, 250 Thevon Street, GR-12241 Athens, Greece; pgfragouli@uniwa.gr

**Keywords:** polypeptides, ring-opening polymerization, high vacuum techniques, *N*-carboxy anhydrides, protective groups, orthogonal system, stimuli-responsive polypeptides

## Abstract

The synthesis of well-defined polypeptides exhibiting complex macromolecular architectures requires the use of monomers that can be orthogonally deprotected, containing primary amines that will be used as the initiator for the Ring Opening Polymerization (ROP) of N-carboxy anhydrides. The synthesis and characterization of the novel monomer *N^ε^*-9-Fluorenylmethoxycarbonyl-l-Lysine *N*-carboxy anhydride (*N^ε^*-Fmoc-l-Lysine NCA), as well as the novel linear Poly(*N^ε^*-Fmoc-l-Lys)_n_ homopolypeptide and Poly(l-Lysine)_78_–*block*–[Poly(l-Lysine)_10_–*graf*t–Poly(l-Histidine)_15_] block-graft copolypeptide, are presented. The synthesis of the graft copolypeptide was conducted via ROP of the *N^ε^*-Boc-l-Lysine NCA while using *n*-hexylamine as the initiator, followed by the polymerization of *N^ε^*-Fmoc-l-Lysine NCA. The last block was selectively deprotected under basic conditions, and the resulting *ε*-amines were used as the initiating species for the ROP of *N^im^*-Trityl-l-Histidine NCA. Finally, the *Boc*- and *Trt*- groups were deprotected by TFA. High Vacuum Techniques were applied to achieve the conditions that are required for the synthesis of well-defined polypeptides. The molecular characterization indicated that the polypeptides exhibited high degree of molecular and compositional homogeneity. Finally, Dynamic Light Scattering, ζ-potential, and Circular Dichroism measurements were used in order to investigate the ability of the polypeptide to self-assemble in different conditions. This monomer opens avenues for the synthesis of polypeptides with complex macromolecular architectures that can define the aggregation behavior, and, therefore, can lead to the synthesis of “smart” stimuli-responsive nanocarriers for controlled drug delivery applications.

## 1. Introduction

Currently there is great interest, both academic and industrial, in the synthesis of biocompatible and biodegradable polymers, and polypeptides are the main materials of this class. Proteins are the most widely used materials in nature that perform the most sophisticated mode of actions through their exceptional ability to self-assemble into complex three-dimensional structures [1,2]. Polypeptides present the advantages of natural proteins, such as secondary and tertiary structures, and they are used in many bio-related applications, e.g., nanomedicine, which focus on the effective encapsulation of pharmaceutical compounds and control of their biodistribution [3,4,5].

Recently, amphiphilic polypeptides, rather than conventional synthetic polymers, have become one of the most promising materials in the design of drug delivery systems (DDS). Their unique secondary structure plays a major role in the self-assembly process and it affects the formation and the properties of the final nanostructures [5,6,7,8,9,10,11,12,13,14]. Branched amphiphilic polypeptides do not activate the immune system response and, therefore, can stay longer in blood circulation, with the obvious advantage of potential improved accumulation to the targeted pathological sites. Branched polymers, such as miktoarm star polymers, which were presented in the early 1990s by our group, were found to control the microphase morphology and volume fraction separately in bulk and significantly influence the aggregation in solution [15].

Orthogonal protective groups are required Ii order to synthesize polypeptides with complex macromolecular architecture through primary amine initiation [16,17]. Such groups allow for the partial selective deprotection of a polypeptide-bearing protected primary amine, such as Lysine, followed by the polymerization of another NCA to form a branched polypeptide. Lysine has been used for the synthesis of hyperbranched polypeptides, through the utilization of the *ε*-amine as the secondary initiation species after the first polymerization step and its deprotection. Several protective groups have been used for the protection of the *ε*-amine of Lysine, including carboxybenzyl (*Z*), *tert*-butoxycarbonyl (*Boc*), and trifluoroacetyl (*TFA*) groups. However, the employment of a pair of these groups as orthogonal protectors to perform the synthesis of branched copolypeptides has not been reported, to our knowledge.

Solid phase peptide synthesis (SPPS) relies on the utilization of appropriate protecting groups, as the chain is being developed without affecting other functional groups. These groups require the orthogonal deprotection, i.e., the selective deprotection of one group without affecting the other. One of the most widely used pairs for the orthogonal protection of a primary amine is the 9-fluorenylmethoxycarbonyl (*Fmoc*) and Benzyloxycarbonyl (*Boc*) pair. Introduced in the 1970s, *Fmoc* protectors are stable in acidic conditions, but sensitive to aqueous base and secondary amines, while *Boc*- is removed under acidic conditions [16,17]. The TFA group has been used for the protection of the *ε*-amine of Lysine, which can be deprotected under mild basic conditions, i.e., by treating with an ammonia solution in a mixture of methanol/water. Because the initiator that we use is a primary amine that can act as a strong base, there is the possibility that some of the TFA groups will be cleaved during the initiation that would lead to the formation of branched materials. The indication of the formation of branched side products in the polymerization of TFA-protected Lysine is found in the work of Rodrıguez-Hernandez et al. [18] as well as of Collet et al. [19]. In both cases, high molecular weight side products were produced. In addition, the cleavage occurs under aqueous conditions, which cannot be applied if the peptide that is to be deprotected contains an additional hydrophobic block. Therefore, in order to exclude such side reactions and perform the deprotection in organic solvent, a more stable group that can be deprotected under more rigorous basic conditions is required, such as the *Fmoc* group.

In this work, the synthesis and characterization of a novel NCA, *N^ε^*-fluorenylmethoxycarbonyl-l-Lysine NCA (*N^ε^*-Fmoc-l-Lysine NCA), is presented. The successful synthesis of this monomer (*N^ε^*-Fmoc-l-Lysine NCA) was evaluated through its characterization by using Nuclear Magnetic Resonance Spectroscopy (^1^H-NMR) and Fourier Transform-Infrared (FT-IR) spectroscopy. This novel NCA was used for the synthesis of two well-defined linear poly(*N^ε^*-Fmoc-l-Lysine) homopolypeptides with different molecular weights while using the ROP of ΝCA and *n*-hexylamine as initiator, followed by the deprotection of the *Fmoc* protective groups with piperidine to form poly(l-Lysine) [17,20]. The protected and deprotected homopolypeptides Poly(*N^ε^*-Fmoc-l-Lys) (PFLL) and Poly(l-Lys) (PLL), respectively, were characterized by ^1^H-NMR, FT-IR, and Size Exclusion Chromatography (SEC) measurements.

In addition, the well-defined block-graft copolypeptide of the type PLL_78_–*b*–(PLL_10_–*g*–PHIS_15_) was synthesized, where PHIS is poly(l-Histidine). The synthetic approach first involved the synthesis of poly(*N^ε^*-Boc-l-Lysine)–*b*–poly(*N^ε^*-Fmoc-l-Lysine) diblock copolypeptide by the sequential ROP of *N^ε^*-Boc-l-Lysine and *N^ε^*-Fmoc-l-Lysine NCAs, while using *n*-hexylamine as initiator, followed by the selective deprotection of *Fmoc* groups with piperidine. This copolypeptide constituted the backbone of the final block-graft copolypeptide. The free amino groups of the second block were used as the initiating species for the ROP of *N^im^*-Trt-l-HIS-NCA, leading to poly(*N^ε^*-Boc-l-Lysine)–*b*–[poly(l-Lysine)–*g*–poly(*N^im^*-Trt-l-Histidine)]. Finally, the *Boc* and *Trt* groups were cleaved by trifluoroacetic acid. ^1^H-NMR, FT-IR, and SEC measurements confirmed the successful synthesis of PLL–*b*–[PLL–*g*–PHIS]. Z-potential, Dynamic Light Scattering (DLS), as well as Circular Dichroism (CD) measurements were also conducted in order to study the polymer’s aggregation behavior and its ability to respond upon changing pH values.

## 2. Materials and Methods

### 2.1. Materials

Ethyl acetate (EtOAc, >99.5%, Merck Millipore, Darmstadt, Germany) was fractionally distilled over phosphorous pentoxide. *N*,*N*-Dimethylformamide (DMF) (99.9+%), special grade for peptide synthesis (Fischer Scientific, Waltham, MA, USA), was further purified by short-path fractional distillation under HV in a custom-made apparatus. The middle fraction was always used. The purified solvent was stored in a flask under vacuum at 3 °C. The purification of Tetrahydrofuran (THF) (anhydrous, max 0.005% water, Merck Millipore) was performed in a HVL. At first, THF was stirred over calcium hydride for one day. Subsequently, THF was distilled under HV in a flask that contains sodium pellets and stirred overnight. Finally, THF was distilled in a flask with sodium-potassium alloy and then stored under vacuum at room temperature. *n*-Hexylamine (≥99%, Merck Millipore) was dried over calcium hydride for one day and then fractionally distilled under HV. Piperidine (99%, Sigma-Aldrich, Saint Louis, MO, USA) was distilled under high vacuum. Triethylamine (Et_3_N, >99%, Acros Organics, Waltham, MA, USA) was dried over calcium hydride overnight and then distilled and stored under vacuum over sodium. The appropriate quantity needed was freshly distilled in a vacuum line prior to use. *n*-Hexane (>99%, Merck Millipore) was dried over calcium hydride overnight, followed by distillation over *n*–butyllithium (*n*–BuLi), while using high vacuum techniques and standard procedures that are required for polymerization. Triphosgene (99%) was purchased from Acros Organics. *N^α^*,*N^ε^*-di-(*tert*-butyloxycarbonyl)-l-Lysine (98%) and *N^ε^*-Fmoc-*N^α^*-Boc-l-Lysine (98%) were purchased from Alfa Aesar (Waltham, MA, USA). Boc-HIS(Trt)–OH was purchased from Christof Senn Laboratories AG (Dielsdorf, Switzerland). All of the other solvents and reagents were purchased from commercial suppliers and used as received.

### 2.2. Physical and Analytical Methods

^1^H-NMR (400 MHz) was performed while using a Bruker 400 spectrometer (Billerica, MA, USA). The spectra of the polypeptides and NCAs were obtained either in D_2_O (polypeptides), deuterium trifluoroacetic acid (*d*-TFA) (polypeptides), or CDCl_3_ (NCAs), at room temperature. FT-IR measurements were performed utilizing a Perkin Elmer Spectrum One instrument (Waltham, MA, USA), in KBr pellets at room temperature, in the range of 450–4000 cm^−1^. Size Exclusion Chromatography (SEC) analysis was performed using two SEC sets. The one was composed of a Waters Breeze instrument (Milford, MA, USA) that was equipped with a 2410 differential refractometer and a Precision PD 2020 two angles (15°, 90°) light scattering detector. The carrier solvent was a 0.10% TFA (*v/v*) solution of water/acetonitrile (60/40 *v/v*) at a flow rate of 0.8 mL min^−1^ at 35 °C. The stationary phase consisted of three linear Waters hydrogel columns. A second SEC instrument was used for the analysis of protected copolypeptides. The system was composed of a Waters 600 HPLC pump, Waters Ultrastyragel columns (HT-2, HT-4, HT-5E, and HT-6E), a Waters 410 differential refractometer, and a Precision PD 2020 two angles (15°, 90°) light scattering detector at 60 °C. The carrier solvent used was a solution of 0.1 M LiBr in DMF with a flow rate of 1 mL min^−1^. Dynamic Light Scattering (DLS) measurements were conducted with a Brookhaven Instruments (Holtsville, NY, USA) Bl-200SM Research Goniometer System (Holtsville, NY, USA) operating at λ = 640 nm and with 40 mW power. Correlation functions were analyzed by the cumulant method and Contin software (Holtsville, NY, USA). The correlation function was collected at 45, 90, and 135°, at 25 °C. All of the measurements were performed in either an isotonic PBS buffer (10 mM, 150 mM NaCl) at pH = 7.4, PBS buffer (10 mM, 150 mM NaCl) at pH = 6.5, or an isotonic acetate buffer (10 mM, 150 mM NaCl) at pH = 5.0. The concentration range measured was between 2.0 × 10^−3^–1.0 × 10^−5^ g mL^−1^. A JASCO J–815 model (Jasco Corporation, Tokyo, Japan) was used in order to obtain the Circular Dichroism spectrum. A 0.1 cm cell well filled with polypeptidic aqueous solutions at a concentration close to 1.0 × 10^−4^ g mL^−1^ was also used. The pH adjustment was achieved by Milli–Q water and the subsequent addition of 0.01 N aqueous HCl droplets in order to gradually lower the pH and obtain the corresponding spectrum at each intermediate value. The pH was adjusted while using a digital pH meter. Similarly, a gradual increase in pH was achieved by adding droplets of 0.01 N aqueous solution of NaOH. The temperature was set to 25 °C using a dedicated digital thermostat. The nitrogen flow was adjusted to 6.0 L min^−1^. Z-potential measurements of the nanoparticles was conducted by using a Brookhaven Instruments Nanobrook Omni (Holtsville, NY, USA) system operating at λ = 640 nm and with 40 mW power, with the Smoluchowski method, operating in Electrophoretic Light Scattering (ELS) mode. The measurements were performed at a neutral pH value.

### 2.3. High Vacuum Techniques

High vacuum techniques were used for the synthesis of desired monomers and polypeptides and for the purification of the reagents and solvents [21,22,23]. The high vacuum technique contributes to the removal of possible impurities from reagents, such as air and moisture, which leads to the undesirable initiation of polymerization. It is also possible to remove impurities, such as amines, alcohols, acids, or stabilizers contained in commercially available solvents and that have the ability to act either as initiators in ROP of *N*-carboxy anhydrides or causing premature termination.

### 2.4. Synthesis and Purification of Monomers (N-Carboxy Anhydrides of α-Amino Acids)

The monomers that are used in the synthesis of well-defined polypeptides (*N*-Carboxy Anhydrides of *α*-amino acids, NCAs) are not commercially available (mostly) and can only be synthesized in the laboratory [21,22,23]. The methods of synthesis and purification of the *N^ε^*-Boc-l-Lys NCA, *N^ε^*-Fmoc-l-Lys NCA, and *N^im^*-Trt-l-HIS NCA monomers are described below.

#### 2.4.1. Synthesis of *N^ε^*-9-Fluorenylmethoxycarbonyl-l-Lysine NCA (*N^ε^*-Fmoc-l-Lys NCA)

Initially, 7.6 g of the precursor compound *N^ε^*-Fmoc-*N^α^*-Βoc-l-Lys (16.22 mmol, M_n_ = 468.55 g mol^−1^) were placed in a 500 mL two-neck round-bottom flask containing a magnetic stirrer and connected to the high vacuum line, and the solid was left to dry overnight. The next day a neighboring round-bottom flask containing pure THF was connected to the high vacuum line. Flame-drying was carried out in the line in order to remove traces of moisture and solvents, and immediately 150 mL of THF were distilled into the flask with the precursor compound. After complete dissolution of the solid precursor, the reactor was transferred to the fume hood. A suitable pad for the continuous supply of argon and bubbler was installed at the flask to control the flow of argon, while a glass dropping funnel with a stopcock was placed at the second entrance of the flask, through which the reagents will be added (Appendix A). Triphosgene (2.41 g, 8.11 mmol, M_n_ = 296.7 g mol^−1^) was added to the 50 mL flask containing pure THF under continuous argon flow. After the complete dissolution of triphosgene, the contents were poured into the dropping funnel. The addition of triphosgene with THF was made dropwise over a period of 30 min under continuous stirring at room temperature. After complete addition, the dropping funnel was removed from the device, the reactor was sealed, and the reaction was kept stirring continuously for another 30 min. At the same time, a calibrated ampoule for the collection of triethylamine (Et_3_N) and a 50 mL round-bottom flask with a stopcock adapter were connected to the high vacuum line. After extensive flame-drying, 2.26 mL Et_3_N were distilled into the ampoule (16.22 mmol, M_n_ = 101.19 g mol^−1^, d = 0.726 g mL^−1^), and approximately 30 mL of pure THF were also distilled in the 50 mL flask. At the same time, a dropping funnel with a stopcock was fitted to the reaction device and the reactor was placed in an ice bath. Finally, Et_3_N was poured in the dropping funnel under a slight flow of argon. The dilution of triethylamine is an important step in avoiding local excess. The diluted Et_3_N was added dropwise to the reaction product over a period of 30 min under intense stirring. After the completion of the addition, the dropping funnel was removed from the device, the flask was sealed, and the reaction was maintained for 5 h in an ice bath, under vigorous stirring and continuous flow of argon. The reaction was monitored by FT-IR. After 5 h, the formation of triethylamine salt as a white sediment was observed. Vacuum filtration was followed under argon flow while using a por.3 glass filter to remove triethylamine salt and the filtrate was collected in a 500 mL two-necked round-bottom flask. The reactor was then attached to the high vacuum line and the solvent was removed until dryness. The resulting solid, which has the form of foam, was left to dry overnight on the high vacuum line.

The purification of the monomer took place by recrystallization with solvents while using ethyl acetate and hexane. Firstly, 200 mL of pure EtOAc were distilled into the reactor containing the monomer. Traces of triethylamine hydrochloride (Et_3_N·HCl) are insoluble in EtOAc (sediment formation), while *N^ε^*-Fmoc-l-Lys NCA remains soluble. Therefore, the separation of the two compounds was carried out by vacuum filtration into with a por3 filter. The filtrate, containing the pure *N^ε^*-Fmoc-l-Lys NCA, was dried with MgSO_4_, filtered, and immediately poured into ~2 L of hexane where the pure monomer was submerged, continuously under an argon flow. Subsequently, the flask filled with argon was placed at –20 °C until the next day, when it was filtered under vacuum under continuous flow of argon in a two-neck 2 L round-bottom flask via a glass filter por. 4, where *N^ε^*-Fmoc-l-Lys ΝCA remains in the filter. The solid was collected in a 250 mL round-bottom flask with a suitable adapter and then left to dryness overnight. The recrystallization process was repeated the following day, in order to achieve the high purity of the monomer, and the product was left in the high vacuum line. The next day, the flask was removed from the high vacuum line and it was stored in a glove box (5.03 g, 12.75 mmol). The purity of *N^ε^*-Fmoc-l-Lys NCA was confirmed by ^1^H-NMR and FT-IR spectroscopy. Scheme 1 summarizes the course of reactions followed for the synthesis of *N^ε^*-Fmoc-l-Lys NCA. The typical yield was 78.6%.

#### 2.4.2. Synthesis of *N^ε^*-tert-butyloxycarbonyl-l-Lysine *N*-Carboxy Anhydride (*N^ε^*-Boc-l-Lys NCA)

Briefly, *N^α^*,*N^ε^*-di-(*tert*-butyloxycarbonyl)-l-Lysine was added to a flask, placed on the vacuum line, and then pumped overnight. Purified ethyl acetate was then distilled, followed by argon insertion in order to reach atmospheric pressure, and then by the addition of triphosgene. The mixture was left to react for 10 min. Triethylamine that was diluted in dry ethyl acetate was subsequently added dropwise, and the solution was immersed in an ice-water bath for 2.5 h. The precipitate was filtered, in order to remove the HCl salt of triethylamine, and the organic filtrate was washed with 6% NaCl until pH = 4 of the aqueous phase was achieved. The organic phase was washed with 0.5% NaHCO_3_ (aqueous) until a neutral pH of the aqueous phase was achieved. The organic phase was separated, dried over MgSO_4_, filtered, and then concentrated to approximately ^1^/_3_ of the initial volume. The addition of hexane resulted in the precipitation of the desired product, which was filtered, dried, and recrystallized two more times from EtOAc/hexane. Finally, *N^ε^*-Boc-l-Lys NCA was dried under high vacuum overnight and then transferred into a glove box [24]. The purity of *N^ε^*-Boc-l-Lys NCA was confirmed by ^1^H-NMR (Appendix A) and FT-IR spectroscopy (Appendix A). Appendix A shows the reactions used for the synthesis of *N^ε^*-Boc-l-Lys NCA. The typical yield was 40.3%.

#### 2.4.3. Synthesis of *N^im^*-trityl-l-Histidine *N*-Carboxy Anhydride (*N^im^*-Trt-HIS-NCA)

*N^im^*-trityl-l-Histidine NCA was synthesized according to a previously reported method [25]. The synthesis of *N^im^*-Trt-HIS-NCA was performed in two steps. In the first step, the HCl salt of *N^im^*-Trt-HIS-NCA was synthesized, followed by the removal of the HCl to afford the pure *N^im^*-Trt-HIS-NCA monomer.

Briefly, Boc-HIS(Trt)–OH was added to a flask and then dried overnight under HV. Purified tetrahydrofuran was then distilled in the flask, followed by argon insertion in order to reach atmospheric pressure and by the addition of thionyl chloride. After 2.5 h, the solution was poured into cold diethyl ether with precipitation of *N^im^*-Trt-HIS-NCA·HCl as the major product. The solid was filtered and dried under HV. The product was recrystallized by distilling ethyl acetate under HV. The flask was placed in a water bath at 45 °C for 1 h, which resulted in dissolution, and then cooled to 0 °C with an ice bath. As a result, the hydrochloride salt of Trt-HIS NCA was precipitated, then filtered, and dried overnight under HV (Appendix A).

Subsequently, purified ethyl acetate was distilled into the flask filled with argon and it was placed in an ice bath. Stoichiometric amount of triethylamine dissolved in ethyl acetate was slowly added dropwise. The resulting triethylamine hydrochloride was filtered off, and the filtrate was poured into hexane in order to recrystallize the *N^im^*-Trt-HIS NCA. The *N^im^*-Trt-HIS-NCA precipitated as a white solid and it was isolated by filtration, was dried under HV overnight, and then transferred into a glove box. The purity of *N^im^*-Trt-HIS-NCA was confirmed by ^1^H-NMR (Appendix A) and FT-IR spectroscopy. Appendix A shows the reactions employed in the synthesis of N^im^-Trt-HIS-NCA. The typical yield was 58%.

### 2.5. Synthesis of Poly(N^ε^-Fmoc-l-Lys) Homopolypeptide

After the synthesis of the novel and highly pure *N^ε^*-Fmoc-l-Lys NCA, we proceeded with its polymerization with *n*-hexylamine as initiator in order to verify that the purity of the monomer is sufficient for its controlled ROP. Even traces of impurities not observed by ^1^H NMR spectroscopy could lead to termination reactions. Therefore, two homopolypeptides with molecular weights of M_n_ = 3.25 × 10^3^ g mol^−1^ and 10.0 × 10^3^ g mol^−1^, respectively, were synthesized while using *n*-hexylamine as the initiator, followed by the removal of *Fmoc* groups, resulting in the homopolypeptide poly(l-Lys)_25_ and poly(l-Lys)_78_, respectively. The same experimental procedure was followed for the synthesis of both copolypeptides and, for this reason, only the synthesis of homopolypeptide poly(*N^ε^*-Fmoc-l-Lys)_25_ will be described in detail. The polymerization lasted three days, while for Poly(*N^ε^*-Fmoc-l-Lys)_78_, t lasted four days.

A special polymerization reactor (see Appendix A) was custom-made, including a 250 mL polymerization flask (A) and two ampoules with break–seals. The first ampoule (B) included the *n*-hexylamine (0.06 mmol), and the second one (C) was used for the addition of the monomer. The custom–made glass apparatus was attached to the high–vacuum line through the ground joint (D) in order to be evacuated and flame-dried several times. Subsequently, almost 10 mL of the polymerization solvent—purified *N*,*N*-dimethylformamide—were distilled into the polymerization flask. The glass apparatus was subsequently disconnected from the high–vacuum line and then reattached via the side ampoule (E) in order to remove humidity by flame–drying. The reactor was then transferred to the glove box, where 1.86 g of *N^ε^*-Fmoc-l-Lys NCA (4.71 mmol, Μ_n_ = 394.4 g mol^−1^) were added to the ampoule under argon. The ampoule was subsequently connected to the high vacuum line (E) and it was degassed in order to remove the argon. Subsequently, 8 mL of pure DMF were distilled into the ampoule with the monomer. The device was removed from the line via heat sealing. The monomer was completely dissolved in the DMF and, by rupturing the break–seal of the ampoule, the NCA solution was added to the solution of the flask. The break–seal of the ampoule with the initiator was subsequently ruptured, and the contents of the solution were poured quantitatively and stirred continuously in the polymerization flask in order to initiate the polymerization. The ROP resulted in the release of carbon dioxide, which was an indication of the success of the initiation and propagation of the polymerization until completion. A sample was collected from the flask in the glove box for FT-IR spectrum measurement in order to ensure polymerization completion. The absence of the characteristic peaks of the monomer at 1786 and 1821 cm^−1^ as well as the development of the peak at 1650 cm^−1^ due to the peptide bond revealed the completion of the reaction. The polymerization process lasted 3 days. The protected homopolymer was precipitated in cold diethyl ether, and the solid was then isolated by vacuum filtration in a Buchner funnel while using a 0.45 μm PTFE hydrophobic filter. The white solid collected was transferred to a 50 mL round-bottom flask and left to dry in HVL overnight. Extensive molecular characterization studies were conducted to confirm the successful synthesis of the polymer.

Scheme 2 shows the reactions used for the synthesis of poly(*N^ε^*-Fmoc-l-Lys).

### 2.6. Deprotection Procedure of Poly(Ν^ε^-Fmoc-l-Lys)_n_ Homopolypeptide

To a 50 mL round-bottom flask containing 0.2 g of fully protected polymer, 10 mL of DMF were added in order to dissolve the polymer. Distilled piperidine (10 mL) was then added, which reacted with the *Fmoc* protective groups, forming a yellow solution after intense stirring. The resulting bright yellow, clear solution was left under stirring for 40 min at room temperature. Subsequently, 20 mL of cold water were added in order to precipitate the *Fmoc* protective groups. *Fmoc* side products were then separated from the dissolved polymer by vacuum filtration into a Buchner funnel with a por3 glass filter. The filtrate was transferred to a flask that was connected to the high vacuum line in order to remove the solvents to a neighboring flask, and it was left to dry overnight. The solid was then dissolved in 20 mL Milli–Q water, and the procedure continued with the consequential dialysis of the solution against 2 L of Milli–Q water twice while using a dialysis membrane with MW cutoff of 1000 Daltons. The pH was adjusted to ~7.4 with a dilute aqueous solution of HCl 1 M at room temperature. The dialysis was followed against 2 L of Milli–Q water twice, adjusting the pH to ~7–8 with an aqueous solution of NaOH 1 N. When the pH value remained at 7–8, the membrane with the polymer was added to 2 L of pure Milli–Q water twice. Finally, the solution in the membrane was selected and freeze-dried in order to obtain the product. The fully deprotected polymer was weighed and stored. In summary, the steps that are followed for the deprotection of poly(l-Lysine) blocks are given in Scheme 3. The two deprotected homopolypeptides were also characterized by SEC, ^1^H-NMR and FT-IR spectroscopy (Appendix A).

### 2.7. Synthesis of Poly(l-Lysine)_78_–b–(Poly(l-Lysine)_10_–g–Poly(l-Histidine)_15_) (PLL_78_–b–(PLL_10_–g–PHIS_15_)) Block–Graft Polypeptide

The synthetic approach first involved the synthesis of the backbone, i.e., the synthesis of Poly(*N^ε^*-Boc-l-Lysine)–*block*–Poly(*N^ε^*-Fmoc-l-Lysine) (PBLL_78_–*b*–PFLL_10_) diblock copolypeptide. The copolypeptide was synthesized while using *n*-hexylamine as the initiator, and the sequential addition of the *N^ε^*-Boc-l-Lys NCA (1.27 g, 4.66 mmol), followed, after the completion of polymerization, by the addition of *N^ε^*-Fmoc-l-Lys NCA (0.237 g, 0.60 mmol). The polypeptide was precipitated in diethyl ether, isolated by filtration, and then dried under HV overnight (Scheme 4).

PΒLL_78_–*b*–PLL_10_ copolypeptide was produced by the selective deprotection of the *ε*-amine groups of *Ν^ε^*-Fmoc-l-Lysine. The resulting amine groups of PLL served as the macroinitiator for the ROP of N^im^-Trt-HIS-NCA, in order to afford the *block-graft* copolypeptide.

### 2.8. Selective Cleavage of Fmoc Groups

The *Fmoc* groups were cleaved after the addition of piperidine (30% (*v/v*)) to the solution of the protected copolypeptide in DMF for 1h at 25 °C. The reaction product was subsequently precipitated in excess diethyl ether, filtered, and dried under vacuum. The polymer was dissolved in 10 mL Milli–Q, a few drops of NaOH 1 N were added to bring the pH = 10, and the solution was dialyzed twice against a solution of 2 Liters Milli–Q water with a few drops of NaOH 1 N to reach a pH = 10. The dialysis bag was then placed twice in 2 Liters of pure Milli–Q water to remove the salts and it was subsequently freeze-dried, in order to afford 0.374 g of PBLL_78_–*b*–PLL_10_ (Scheme 5).

### 2.9. Synthesis of PLL_78_–b–(PLL_10_–g-PHIS_15_) Block–Graft Polypeptide

The resulting PΒLL_78_–*b*–PLL_10_ diblock was dissolved in purified DMF, followed by the addition of 1.27 g (3.0 mmol) of *N^im^*-Trt-HIS-NCA. The polymerization was left to completion for five days, followed by precipitation in diethyl ether. The *block*–*graft* polypeptide was isolated by filtration and then dried under HV overnight (Scheme 6).

### 2.10. Cleavage of Boc and Trt groups

The *Boc* and *Trt* groups were removed by treating the protected polypeptide with TFA for 60 min at room temperature. Excess (Et)_3_SiH, which acts as a scavenger of the free trityl groups, was added dropwise. The polymer was precipitated by adding the reaction mixture to an excess of diethyl ether, and it was isolated by filtration and dried under vacuum.

The white solid was dissolved in 15 mL of Milli–Q at a pH~5, which was adjusted by the addition of dilute Κ_2_CO_3_ and dialyzed against 2 Liters of Milli–Q water with pH~4 (adjusted with a dilute aqueous solution of HCl) two times against 2 Liters of Milli–Q water with pH~10 (adjusted with a dilute aqueous solution of NaOH) and two times with pure Milli–Q water. Finally, the polymer solution was freeze-dried (Scheme 7). The polymer was characterized by SEC, ^1^H-NMR, and FT-IR spectroscopy.

### 2.11. Formation of the Aggregates

A solution of the polymer in DMSO was initially prepared in order to form the aggregates. Specifically, 2 mg of the polymer were dissolved in 1 mL DMSO and left under stirring overnight. The next day, 6 mL of the PBS buffer (10 mM, 150 mM NaCl) with pH = 7.4, were added to the solution of the polymer. The aggregation of the amphiphilic block-graft copolypeptides occurred by this solvent–switch methodology from the good solvent DMSO into aqueous solutions. The solution was then transferred to a dialysis membrane (MWCO = 3.5 k), and the procedure was performed by adjusting the pH to 7.4. The membrane was dialyzed three times, every three hours, and finally, the solution was left overnight for dialysis.

The samples were filtered with a 0.45 μm hydrophilic filter prior to the DLS measurements. The solution was subsequently divided into two parts, and each was transferred to a dialysis membrane (MWCO = 3.5 k). The dialysis procedure was performed by adjusting the pH value to 6.5 and 5.0, respectively. For pH values pf 6.5, a PBS buffer solution (10 mM, 150 mM NaCl) was used, while, for pH value pf 5.0, an acetate buffer solution (10 mM, 150 mM NaCl) was used. After three changes of buffers every three hours, the solutions were left overnight and they were analyzed by dynamic light scattering the next day.

## 3. Results and Discussion

### 3.1. Synthesis and Characterization of the N^ε^-Fmoc-l-Lysine NCA

FT-IR spectroscopy monitored the synthesis of the *N^ε^*-Fmoc-l-Lysine NCA. The peak at 1754 cm^−1^ is attributed to the carbonyl group of the *Boc* group, while the peaks at 745, 759 cm^−1^, and 1706 cm^−1^ are due to the aromatic (the first two) and the carbonyl groups (the last) of the *Fmoc* group. The peak at 1653 cm^−1^ corresponds to the vibration of the carbonyl group (C=O) of the carboxylic acid group of the precursor. Triphosgene was used as a chlorination agent in order to activate the *Boc* group, followed by the subsequent addition of triethylamine to remove the hydrochloride produced during the reaction through precipitation. The monitoring of the reaction was confirmed by FT-IR. In the final spectrum, *N^ε^*-Fmoc-l-Lys NCA, the characteristic peak at 1653 and 1754 cm^−1^ is absent, confirming the anhydride formation.

The peaks at approximately 743, 759 cm^−1^ (vibrations of the bonds –CH=CH– of benzene rings) and 1697 cm^−1^ corresponding to the *Fmoc* protective group remain, indicating that the protective group of the monomer remained intact throughout the synthetic procedure. The appearance of the two characteristic peaks at 1786 (C2) and 1859 cm^−1^ (C5) that correspond to the symmetrical and asymmetric vibration of the anhydride carbonyls indicates the successful synthesis of *N*-carboxy anhydride (Figure 1).

The precursor and the final purified *N^ε^*-Fmoc-l-Lysine NCA was characterized by ^1^H-NMR spectroscopy in CDCl_3_. Figure 2 provides the spectrum of the precursor. ^1^H-NMR (400 MHz, CDCl_3_, δ, ppm): 1.0–1.6 (9H from the Boc- group, 4H from –R group, –HN–CH_2_–CH_2_–CH_2_–CH_2_–), 1.71–1.86 (2H, from –R group, –HN–CH_2_–CH_2_–CH_2_–CH_2_–), 3.18 (2H, from –R group, –HN–CH_2_–CH_2_–CH_2_–CH_2_–), 4.20 (1H, N–CH–CO), 4.31 (1H, –CH– of the five-member ring of the protective group), 4.40 (2H, –NH–CO–O–CH_2_–), 4.91 (1H, –CH_2_–NH–CO– of *ε*-amine), 6.19 (1H, CH–NH–CO), and 7.30–7.74 (8H, ArH of Fmoc protective group).

The spectrum of the final NCA is presented on Figure 3, where the characteristic peaks of the *Boc*- groups at 1.4 ppm are diminished. ^1^H-NMR (400 MHz, CDCl_3_, δ, ppm): 1.45–2.03 (6H from the –R group, –HN–CH_2_–CH_2_–CH_2_–CH_2_–), 3.20 (2H, from the –R group, –HN–CH_2_–CH_2_–CH_2_–CH_2_–), 4.23 (1H, N–CH–CO), 4.30 (1H, –CH– of the five-member ring of the protective group), 4.46 (2H, –NH–CO–O–CH_2_–), 4.93, (1H, –CH_2_–NH–CO– of *ε*-amine), 6.86 (1H, CH–NH–CO of the NCA ring), and 7.33–7.78 (8H, ArH of *Fmoc* protective group). It is evident that the synthesis of the final *N*-carboxy anhydride, *N^ε^*-Fmoc-l-Lys NCA, was successful.

### 3.2. Synthesis and Characterization of Protected and Deprotected Poly(N^ε^-Fmoc-l-Lysine) Homopolypeptides

The completion of the polymerization and the consumption of monomer was monitored by FT-IR spectroscopy. The duration of the polymerization of the low molecular weight Poly(*N^ε^*-Fmoc-l-Lysine) was three days, while of the higher molecular weight required four days. This kinetic is slower than that of the Lysine NCA with *Boc*-protective group, which last 24 and 48 h, respectively. This can be attributed to the bulkier *Fmoc* protective group that hinders the reaction of the NCA with the amine. In addition, the deprotection of the homopolymer was confirmed by FT-IR and ^1^H NMR spectroscopy. Figure 4 shows the FT-IR spectra of both protected and deprotected homopolypeptides. The deprotection of the homopolypeptides is revealed, as expected in the FT-IR spectrum of poly(l-Lysine). The peak at 1657 cm^−1^ is attributed to the carbonyl (C=O) of the amide bond formed through the polymerization. Furthermore, the disappearance of the peaks at 741, 761, and 1706 cm^−1^ indicates the complete cleavage of the *Fmoc* group.

The ^1^H-NMR spectra of the protected and deprotected PLL proved the complete cleavage of the *Fmoc* group. Figure 5 provides the ^1^H-NMR spectrum of Poly(*N^ε^*-Fmoc-l-Lysine)_78_: (400 MHz, CDCl_3_, δ, ppm): 1.52 (4H, –CH_2_–CH_2_–CH_2_–CH_2_–NH–), 1.97 (2H, –CH_2_–CH_2_–CH_2_–CH_2_–NH–), 3.12 (2H, –CH_2_–), 3.98 (2H, –CH_2_–CH_2_–CH_2_–CH_2_–NH–), 4.22 (2H, –CH_2_–O–C=O–), 5.55 (1H, –NH–, backbone), 7.11–7.61 (8H, =C–H, ArH of the Fmoc group), and 8.32 (1H, –CH–NH–H–). FT-IR (KBr pellet): 1651 cm^−1^ (v CO, amide bond), 741, 763 cm^−1^ (vibrations of the –CH=CH– of the Fmoc benzene rings), and 1706 cm^−1^ (vibration of the carbonyls –C=O of the *Fmoc* protective groups).

Figure 6 shows the corresponding ^1^H-NMR spectrum of the deprotected homopolymer in D_2_O. The peaks at 7.0–8.0 ppm corresponding to the aromatic hydrogens of *Fmoc* protective groups are almost absent, indicating that deprotection occurred for at least with 99.5% yield. (400 MHz, D_2_O, δ, ppm): 1.34 (2H, –CH_2_–CH_2_–CH_2_–CH_2_–NH_2_), 1.59 (4H, –CH_2_–CH_2_–CH_2_–CH_2_–NH_2_), 2.90 (2H, –CH_2_–CH_2_–CH_2_–CH_2_–NH_2_), and 4.19 (1H, –NH–CH(R)–C=O). FT-IR for Poly(*Ν^ε^*-Fmoc-l-Lys)_78_: 1648 cm^−1^ (v CO, amide bond).

The protected Poly(*N^ε^*-Fmoc-l-Lysine) homopolymers were not soluble in 0.1 M LiBr in DMF at 60 °C. The deprotected polypeptides were also examined by size exclusion chromatography in aqueous SEC (Figure 7). The polydispersity index for Poly(l-Lysine)_25_ and Poly(l-Lysine)_78_ was 1.15 and 1.09, respectively.

### 3.3. Synthesis and Characterization of Poly(l-Lysine)–b–(Poly(l-Lysine)–g–Poly(l-Histidine)) Block–Graft Copolypeptide (PLL–b–(PLL–g–PHIS))

The synthetic strategy for the synthesis of the block-graft copolypeptide first involved the synthesis of the poly(*N^ε^*-Boc-l-Lysine)–*b*–poly(*N^ε^*-FMoc-l-Lysine) (PBLL_78_–*b*–PFLL_10_) diblock copolypeptide, followed by the selective deprotection of the *Fmoc* groups of the second block. The resulting free amino groups are capable of initiating ROP polymerization of the *N*-carboxy anhydride of Histidine. Finally, the groups of Lysine and Histidine were deprotected in order to obtain the desired product.

#### 3.3.1. Synthesis and Characterization of Poly(N^ε^-Boc-l-Lysine)_78_–b–Poly(N^ε^-Fmoc-l-Lysine)_10_ (PBLL_78_–b–PFLL_10_) Diblock Copolypeptide

The duration of the polymerization of the *Boc*-protected NCA was almost the same with that of the *Fmoc* protected monomer, i.e., almost two days, indicating that the polymerization of the *Fmoc* monomer is slower than that of the *Boc*- monomer. FT-IR infrared spectroscopy was employed in order to monitor the synthesis of the PBLL_78_–*b*–PFLL_10_ copolypeptide (Figure 8). The completion of the polymerization of the two blocks can be confirmed by the diminished peaks of the NCAs as well as the development of the peak at 1652 cm^−1^ (Figure 8C) of the amide bond. In addition, the characteristic peaks at 1694 and 1708 cm^−1^, which correspond to the carbonyls (C=O) of the protective groups *Boc* and *Fmoc*, respectively, are shown in spectrum 8C.

#### 3.3.2. Selective Deprotection of the Fmoc Protected Block

The selective removal of the *Fmoc* groups through the procedure mentioned above was carried out. The polymer was characterized by ^1^H-NMR and SEC. The deprotection of the 2nd block bearing the *Fmoc* group is evident. since the peaks of the protons corresponding to the aromatic hydrogens at 7.30–7.80 ppm are not present, while the protons of the *Boc*- group are obvious, as shown in Figure 9. (400 MHz, CDCl_3_, δ, ppm): 1.00-1.7 9H from the *Boc*- group, 6H from –R group, 3.08 (2H, –CH_2_–CH_2_–CH_2_–C**H**_2_–NH_2_), and 5.14 (1H, –NH–C**H**(R)–C=O).

The polymer was further characterized by SEC, as shown in Figure 10.

The SEC eluogram presented in Figure 10 was measured in DMF/LiBr solvent. Although the deprotected PLL block is not completely soluble in this solvent, no aggregation occurred, and a monomodal peak was present. The molecular weight of PBLL_78_–*b*–PLL_10_ obtained was M_n_ = 19.4 × 10^3^ g mol^−1^, while the stoichiometrically expected molecular weight was 18.1 × 10^3^ g mol^−1^. The polydispersity index was 1.17. Scheme 8 depicts the synthesized graft copolypeptides.

#### 3.3.3. Synthesis and Characterization of the Final PLL_78_–b–(PLL_10_–g–PHIS_15_)

The free amino groups of the 2nd block of PBLL_78_–*b*–PLL_10_ were used as the initiators for the ROP of the *N^im^*-Trt-l-HIS NCA. The FT-IR spectra of the protected PBLL_78_–*b*–(PLL_10_–*g*–P(Trt)HIS_15_) show the expected peaks (Figure 11).

No further characterization was carried out at this stage of the procedure, because the protected Histidine blocks were not sufficiently dissolved in order to obtain either an ^1^H-NMR or SEC spectrum.

Figure 12 provides the FT-IR spectrum of the final product. The complete removal of *Trt*- group is verified by the absence of the peaks at 703 and 750 cm^−1^ attributed to the *Trt* group of Histidine, while the absence of the peaks at 1712 and 1380 cm^−1^ indicate the complete cleavage of the *Boc* group (Figure 12B).

In addition, the deprotected polypeptide was subsequently characterized by ^1^H-NMR spectroscopy (Figure 13). ^1^H-NMR (400 MHz, d-TFA, δ, ppm): 1.10–1.60 (6H, –HN–CH_2_–CH_2_–CH_2_–CH_2_–), 2.70–3.10 (2H, –CH_2_–ΙΜ, 2H, –HN–CH_2_–CH_2_–CH_2_–CH_2_–), 4.22 (1H, NH–CH(CH_2_– IΜ)–C=O), 4.67 (1H, –NH–CH(R)–C=O), 7.06 (1H, C=CH–N–), 8.22 (1H, –N=CH–N–), and FT-IR (thin film): 1640 cm^−1^ (v CO, amide I). As observed, the integration of the peaks almost coincides with the theoretically predicted values, thus confirming the successful synthesis.

The further characterization of the final polymer was performed in aqueous SEC (Figure 14). The molecular weight distribution of the PLL_78_–*b*–(PLL_10_–*g*–PHIS_15_) was equal to 1.2, and the obtained molecular weight was M_n_ = 29.7 × 10^3^ g mol^−1^ close to the stoichiometric, equal to 32.0 × 10^3^ g mol^−1^. The low polydispersity index, as well as the agreement of the stoichiometric composition to that experimentally obtained, indicate the high degree of molecular and compositional homogeneity of the block-graft copolypeptide.

### 3.4. Study of the Secondary Structure of PLL_78_–b–(PLL_10_–g–PHIS_15_) Block–Graft Copolypeptide Using Circular Dichroism

The secondary structure of proteins plays a leading role in the transportation and selective release of drugs into the living organism, since proteins have the ability to respond to external stimuli, such as pH. In this polypeptide, only poly(l-Histidine) responds within the physiological pH range (5.0 > pH > 7.4), while PLL remains protonated and it adopts the random coil conformation within this range, due to electrostatic repulsions. The transformation of the secondary structure of the block-graft copolypeptide is due to PHIS, which has pKa = 6.5. In a previous work, we have found that PHIS adopts a β-turn conformation at pH higher than 7.4, while it transforms to random coil by lowering the pH and becoming protonated [26,27,28,29,30].

Figure 15 provides the Circular Dichroism data for the PLL_78_–*b*–(PLL_10_–*g*–PHIS_15_) at pH values 5.0, 6.5, and 7.4. It is obvious that at all pH values, the conformation is mainly random coil due to the contribution of PLL. At pH = 5.0, both of the polypeptides adopt the random coil conformation due to their charged nature. By increasing the pH higher than 5.0, the positive peak at 212 nm attributed to the random coil conformation is lowered, and a new positive peak at 201 nm that is attributed to the β-turn conformation of PHIS appears. Furthermore, the negative peak at 192 nm attributed to the random coil conformation is shifted to 189 nm, due to the formation of the β-turn of the PHIS. 

### 3.5. Self-Assembly Behavior of PLL_78_–b–(PLL_10_–g–PHIS_15_) by Dynamic Light Scattering and ζ-Potential Measurements

The amphiphilic block-graft copolypeptide is expected to self-assemble in aqueous media and respond to pH change. The hydrophilic shell of the formed nanostructures is expected to be comprised of the hydrophilic block PLL at the corona, while the core is based on hydrophobic and pH-sensitive PHIS. The diameter (D) of the aggregates was investigated by DLS and ζ-potential measurements at pH = 7.4 (pH of healthy human tissues), pH = 6.5 (pH in the extracellular region of cancer cells), and pH = 5 (pH of intracellular late endosomes). At each pH, the measurements were conducted at three angles: 45°, 90°, and 135°. All of the measurements were performed at 25 °C (Figure 16).

Table 1 provides the obtained results. The values are the average of three measurements for each solution. The DLS data indicate pH dependence for PLL78–b–(PLL10–g–PHIS15). By reducing the pH value from 7.4 to 6.5, the diameter of the aggregates increases due to the swelling of the core. Thus, an aggregation of the polymer was observed at pH = 7.4, where the polyhistidine chains acted as the hydrophobic moiety, since, at this pH PHIS, is neutral and hydrophobic. As the pH decreased, the diameter of the aggregate started to increase, due to the swelling of its hydrophobic part (at 90°), from 243 nm (pH = 7.4) to 308 (pH = 6.5) and 357 nm (at pH = 5.0). This is due to the gradual protonation of the PHIS grafted blocks. At the same time, a small percentage of a smaller population at 52 nm at pH = 5.0 appears, probably due to degradation of the aggregates and their re-aggregation into smaller nanoparticles due to the protonation of PHIS. Furthermore, the size of the aggregates is angle dependent, which is probably due to the non-spherical shape or to the large dimensions of the aggregates.

### 3.6. Z-Potential Measurement of PLL_78_–b–(PLL_10_–g–PHIS_15_)

In addition, ζ-potential measurements were performed with ELS in a polymer concentration solution of 1 mg mL^−1^, at a neutral pH value, while using the Smoluchowski method (Figure 17).

In this case, the hydrophilic chains of PLL are expected to be at the corona of the aggregates, rendering the Z potential strongly positive and equal to +56.4 mV, due to the positive charges of PLL, which remained at all pH values.

It should be noted that the graft nature of PHIS chains does not allow for a complete disruption of the aggregates at pH = 5.0, as compared to simple linear diblock copolypeptides of the PEO–*b*–PHIS obtained in our previous work. It seems that the grafted nature of PHIS chains renders the core of the aggregate denser, and the PHIS chains cannot be protonated to the same extent and, thus, remain aggregated at pH = 5.0. Probably, a lower pH is required for the complete disruption of the aggregate [25].

## 4. Conclusions

We synthesized the highly pure monomer *N^ε^*-Fmoc-l-Lysine NCA and used it for the synthesis of the corresponding linear homopolypeptides and the block-graft copolypeptide PLL_78_–*b*–(PLL_10_–*g*–PHIS_15_). The characterization results of the polypeptides showed that the polypeptides exhibited a high degree of molecular and compositional (in the case of the copolypeptide) homogeneity. 

The PLL_78_–*b*–(PLL_10_–*g*–PHIS_15_) copolypeptide has the ability to self-assemble in aqueous media and form aggregates with the hydrophilic poly(l-Lysine) that is located at the outer corona and the pH-responsive PHIS at the core. The CD and DLS data exhibited the pH-dependent behavior of PLL_78_–*b*–(PLL_10_–*g*–PHIS_15_) *block-graft* copolypeptide due to pH-responsiveness of PHIS. The CD measurements showed that the graft copolypeptide has the ability to change its secondary structure from random coil conformation to *β*-turn upon increasing the pH value from 5.0 to 7.4. Moreover, the DLS results indicated a swelling of nanoparticles from 243 nm to 357 nm upon decreasing the pH value within the physiological pH range from 7.4 to 5.0. The surface charge of nanoparticles at neutral pH value was found to be equal to +56.4 mV, as expected, due to the presence of poly(l-Lysine) on the outer periphery.

In conclusion, this work opens avenues for the synthesis of multifunctional polypeptides with complex macromolecular architectures. *N*-carboxy anhydrides of Lysine that carry an *ε*-primary amine, which is the initiating species of the NCAs, can be synthesized bearing two different protective groups, *Boc*- or *Fmoc*-, which can be orthogonally deprotected. The *Boc*- group can be selectively deprotected at acidic conditions, while the *Fmoc* deprotects under basic conditions. Therefore, it is possible to create initiating species in a polypeptidic chain and create branched polypeptides, which are expected to have a significant impact on the aggregation behavior of the polypeptides and controlled drug delivery applications.

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
