# Peer review of "Synthesis and Characterization of the Novel Nε-9-Fluorenylmethoxycarbonyl-l-Lysine N-Carboxy Anhydride. Synthesis of Well-Defined Linear and Branched Polypeptides"

_polymers, 2020, doi:10.3390/polym12122819_

Round 1

Reviewer 1 Report

polymers-1001287

The article: “Synthesis and Characterization of the Novel Nε-9-Fluorenylmethoxycarbonyl-L-Lysine N-Carboxy Anhydride. Synthesis of Well-Defined Linear and 4 Branched Polypeptides” by V. Athanasiou et al. describes the synthesis and characterization of the monomer Nε-9 Fluorenylmethoxycarbonyl-L-Lysine N-carboxy anhydride (Nε-Fmoc-L-Lysine NCA), as well  as the linear Poly(Nε-Fmoc-L-Lys)n homopolypeptide and  Poly(L-Lysine)78–block–[Poly(L-Lysine)10–graft–Poly(L-Histidine)15] block-graft copolypeptide. Moreover, the authors examined through DLS, ζ-potential and CD measurements the possibility of self-assembly of these systems.

The MS is well-written and provide to the readers a very detailed experimental description of the procedure followed for the synthesis of the monomer and the corresponding linear and branched copolypeptides. The authors used high vacuum techniques for the preparation and purification of the compounds and the final synthesis of copolypeptides, so the purity of the monomer and the final materials are in high level. I suggest that this work should be published in Polymers. A few comments were raised by reading the manuscript:

  • Figure 7: SEC trace of poly (L-Lysine)78 indicated in blue color seems strange, since the curve is not smooth from both sides (left and right). Maybe the authors can comment on this?
  • Figure 9: I suggest to the authors to assign the peaks with the corresponding protons and provide also the integration values, as they did to all other NMR spectra in the MS.
  • Figure 10: The authors provide the SEC trace of the block copolypeptide after partially deprotection. It’s better to also show the SEC trace of the first as well as the block copolyptide before the deprotection. Did the partially deprotection affect the final SEC trace?
  • The authors provide in some cases the Mn values of the synthesized copolypeptides. For example in page 17, they claim that “The molecular weight of PBLL78-b-PLL10 obtained was Mn=19.4 x 103 g mol-1…” and in page 20, they claim that “PLL78-b-(PLL10-g-PHIS15)…and the molecular weight obtained was Mn=29.7 x 103 g mol-1….” These values are the apparent Mn values that SEC measurements revealed? What samples were used for the calibration of the SEC? The authors should mention that in the instrumentation part.

Author Response

We would like to thank the reviewers for their constructive comments that improved our manuscript. Please find below the point-to-point answers to Reviewer #1

  1. The baseline on blue and red chromatograms are not perfectly flat due to a slight noise probably from the pump, since it is periodic.
  2. We added the integrals. We omitted the integrals at the initial submission due to the fact that the polypeptide was not completely dissolved in chloroform. The protected part was soluble, while the deprotected was not completely soluble and we considered the integration inaccurate. We added this spectrum as a qualitative indication of the removal of the Fmoc- protection indicated by the absence of the peaks at higher than 6.5 ppm.
  3. We wanted also to perform this analysis, it would be very essential for our work. However, the poly(Fmoc-L-lysine) homopolymer as well as the block copolypeptide before deprotection of Fmoc group were not soluble at the SEC instruments we had, i.e. at DMF+LiCl 0.1N at 60 o We added it on lines 516 and 517.
  4. The values obtained were by the light scattering detector, so they are absolute values. We mention that we measured the molecular weights by using the light scattering detectors at the physical and analytical methods section, lines 150-155.

We would like to thank again Reviewer #1 for his consideration.

Reviewer 2 Report

The manuscript under title "Synthesis and characterization of the novel N-9-fluorenylmethoxycarbonyl-L-Lysine N-carboxy anhydride. Synthesis of well defined linear and branched polypeptides" presented to me for review is very interesting. The research topic is very important due to the possibility of using the synthesized peptides in drug delivery systems.

The experiment is well planned and documented. The synthesis of monomers and polymers is multi-step and requires time and precision. This work in question is relevant and is in condition for publication in Polymers. There are no significant problems concerning  this paper. Only some minor complements and corrections are required.

  1. page 7 line 312. There is no mistake made by authors writing that pH after reaction was adjusted by addition of HCl 1M solution?
  2. page 11 line 393. There is no absorption band on IR spectrum of triethylamine HCl salt at wavelength 1653cm-1. Could authors explain the statement that the peak at 1653cm-1 corresponds to this salt?
  3. pages 21-23. Authors write about zeta potential measurements. I think that authors should use one type of symbol, i.e Z or greek letter zeta. The use of uniform symbol facilitates understanding of reading text.
  4. page 11 and 16. Figures 1B and 8B. The spectra of monomer N-Fmoc-L-Lysine NCA little differ on this two graphs. Could authors explain these differences? 

Author Response

We would like to thank the reviewers for their constructive comments. Please find below the point-to-point answers to Reviewer #2.

  1. The aqueous solution after the deprotection of Fmoc groups was basic due to traces of piperidine left. We lowered the pH by adding a few drops of HCl 1N.
  2. The line stated that the peak at 1653 cm-1 was attributed to the HCl salt of triethylamine was removed, it was wrong. It is due to the carboxyl group of amino acid as we mentioned at the same paragraph.
  3. We thank the reviewer, we corrected all terms and stated with the Greek letter ζ.
  4. These two spectra correspond to the same product but are obtained for different batch of NCA. Slight differences of a few cm-1 in the wavelengths are reasonable.

We would like to thank again Reviewer #2 for his consideration.

Reviewer 3 Report

A manuscript by Iatrou and coworkers describes preparation of novel amino acid NCA monomers and their use in synthesis of new homopolypeptides and graft-block copolypeptides via ring opening polymerization. Such polymers have many important uses in numerous fields of applied science and technology, so scientific significance of the work is obvious. The manuscript is clearly written, the experimental part is comprehensively described. In general I support the publication of the manuscript in Polymers after a minor correction. More specific comments are given below:

  • Did the authors try to characterize their polymers by MALDI MS?
  • Line 393: “The peak at 1653 cm–1 corresponds to the HCl salt of triethylamine” – any ref. to support that?
  • Regarding 1H NMR spectra, many of them contain broad or even twinned peaks (e.g., g in Figure 3; e in Figure 2, etc.) due to amide bond rotamers. However, they are described as singlets.
  • Regarding Scheme 9, the description and discussion (line 499) is insufficient. Please describe how the 3D structure was obtained and specify the conditions (media, pH) when the polymer adopts such conformation. How this structure is supported by the experimental data?

Minor points:

  • Line 66: It is better to define abbreviations Z, Boc and TFA as they first appear
  • Line 88: “are presented” -> “is presented”

Author Response

We would like to thank the reviewers for their constructive comments. Please find below the point-to-point answers to Reviewer #3.

  1. Unfortunately, we do not have a MALDI-MS instrument in our Lab nor the Chemistry Department.
  2. Reviewer #2 asked for this also. It was a mistake, we removed this sentence. The peak is due to the carboxyl group of amino acid as we mentioned at the same paragraph.
  3. The peaks of the protons at the amide bonds are wider due to the formation of hydrogen bonding. It is usual for the protons of polypeptides to appear wider than that of small molecules.
  4. Scheme 9 is a qualitative image of the final copolypeptide. It could adopt this conformation at very basic conditions, where the lysine forms α-helices. We used the α-helix conformation for all blocks to show the polypeptidic nature of the polymer.
  5. We inserted the abbreviations of Z, BOC and TFA on line 67.
  6. We corrected line 88.

We would like to thank again Reviewer #3 for his consideration.
